# Empowering Lower Limb Disorder Identification through PoseNet and Artificial Intelligence

**DOI:** 10.3390/diagnostics13182881

**Published:** 2023-09-08

**Authors:** Hafeez Ur Rehman Siddiqui, Adil Ali Saleem, Muhammad Amjad Raza, Santos Gracia Villar, Luis Alonso Dzul Lopez, Isabel de la Torre Diez, Furqan Rustam, Sandra Dudley

**Affiliations:** 1Institute of Computer Science, Khwaja Fareed University of Engineering and Information Technology, Abu Dhabi Road, Rahim Yar Khan 64200, Punjab, Pakistan; hafeez@kfueit.edu.pk (H.U.R.S.); cosc211701003@kfueit.edu.pk (A.A.S.); ch.amjadraza@gmail.com (M.A.R.); 2Universidad Europea del Atlántico, Isabel Torres 21, 39011 Santander, Spain; santos.gracia@uneatlantico.es (S.G.V.); luis.dzul@unini.edu.mx (L.A.D.L.); 3Universidad Internacional Iberoamericana, Campeche 24560, Mexico; 4Department of Extension, Universidade Internacional do Cuanza, Cuito EN250, Bié, Angola; 5Department of Project Management, Universidad Internacional Iberoamericana, Arecibo, PR 00613, USA; 6Department of Signal Theory and Communications and Telematic Engineering, University of Valladolid, Paseo de Belén, 15, 47011 Valladolid, Spain; 7School of Computer Science, University College Dublin, D04 V1W8 Dublin, Ireland; 8Bioengineering Research Centre, School of Engineering, London South Bank University, 103 Borough Road, London SE1 0AA, UK; dudleyms@lsbu.ac.uk

**Keywords:** lower limb disorder, PoseNet, gait analysis, machine learning, Artificial Neural Networks

## Abstract

A novel approach is presented in this study for the classification of lower limb disorders, with a specific emphasis on the knee, hip, and ankle. The research employs gait analysis and the extraction of PoseNet features from video data in order to effectively identify and categorize these disorders. The PoseNet algorithm facilitates the extraction of key body joint movements and positions from videos in a non-invasive and user-friendly manner, thereby offering a comprehensive representation of lower limb movements. The features that are extracted are subsequently standardized and employed as inputs for a range of machine learning algorithms, such as Random Forest, Extra Tree Classifier, Multilayer Perceptron, Artificial Neural Networks, and Convolutional Neural Networks. The models undergo training and testing processes using a dataset consisting of 174 real patients and normal individuals collected at the Tehsil Headquarter Hospital Sadiq Abad. The evaluation of their performance is conducted through the utilization of K-fold cross-validation. The findings exhibit a notable level of accuracy and precision in the classification of various lower limb disorders. Notably, the Artificial Neural Networks model achieves the highest accuracy rate of 98.84%. The proposed methodology exhibits potential in enhancing the diagnosis and treatment planning of lower limb disorders. It presents a non-invasive and efficient method of analyzing gait patterns and identifying particular conditions.

## 1. Introduction

Lower extremity disorders have been identified as a significant factor contributing to disability and reduced quality of life on a global scale [1,2]. Osteoarthritis of the knee, hip, and ankle is a commonly observed disorder affecting the lower limbs [3,4]. These conditions, commonly resulting from trauma, degenerative diseases, or bio-mechanical abnormalities, can give rise to discomfort, a restricted range of motion, and diminished functionality [5,6]. The prompt and precise classification of these conditions is crucial for effective treatment planning, individualized rehabilitation, and the prevention of further consequences. Clinical trials, subjective patient testimonials, and diagnostic imaging modalities such as X-rays and magnetic resonance imaging (MRI) have conventionally served as the predominant approaches in ascertaining the existence and extent of lower limb issues [7,8]. Although these techniques have proven to be valuable, they often require the utilization of specialized equipment, entail significant time investments [9,10], and may not fully capture the comprehensive dynamics of joint movement observed during typical activities. In recent years, advancements in technology have enabled the objective and continuous monitoring of the bio-mechanics in the lower extremities using gait data [11,12].

Gait analysis is employed in a wide variety of domains, including medical diagnostics [13,14,15], osteopathic medicine [16,17], comparative bio-mechanics [18,19,20], and sports-related bio-mechanics [21,22,23]. The application of gait analysis has exhibited considerable promise in the identification and assessment of lower limb disorders [24]. Gait analysis encompasses the evaluation of an individual’s walking pattern, including diverse elements such as stride length, step width, joint angles, and the temporal coordination of movements. Through the examination of these gait parameters, medical professionals and scholars can detect irregularities or deviations from typical gait patterns, which may serve as indicators of the existence of a lower limb disorder [25,26,27]. The objective of the study presented in this manuscript is to investigate the potential application of PoseNet features in the classification of lower limb disorders. PoseNet is a real-time pose estimation model developed by Google, which employs deep learning techniques to accurately assess human poses from both photos and videos [28,29]. Refs. [30,31,32] used PoseNet for in-home rehabilitation, and [33,34] used PoseNet for batsman stroke prediction. These features are capable of capturing the spatial positions and movements of key body joints. PoseNet was chosen by the authors because of its better attributes in the domain of real-time human pose estimation. This is due to its real-time capabilities, user friendliness, and versatility. The model’s ability to perform efficiently with just one video feed, together with its minimal preprocessing requirements and smooth integration into widely used deep learning frameworks, makes it an excellent choice for a wide range of applications. The researchers aimed to establish a dependable and precise classification system for the identification of particular disorders in the hip, ankle, and knee by means of analyzing gait patterns and movements extracted from videos.

PoseNet offers a non-invasive and user-friendly approach to extracting human pose data from videos, obviating the necessity for dedicated apparatus or body-attached markers. This facilitates a more authentic and unimpeded evaluation of gait patterns in real-world contexts. The application of deep learning in PoseNet enables the extraction of complicated and detailed features from videos, resulting in a comprehensive depiction of movements in the lower limbs. The characteristics effectively capture the complex variations and fluctuations in an individual’s gait, which have the potential to serve as indicators for particular disorders affecting the lower limbs. This methodology presents the potential for improved accuracy in diagnosis. The study has the following contributions.

The data were collected from a total of 174 real patients and normal individuals, comprising both male and female participants. The data collection process involved capturing videos of the participants using a camera while they walked on a designated walkway at the Tehsil Headquarter (THQ) Hospital in Sadiqabad.The data were gathered by the system via video recordings, thereby obviating the necessity for intrusive sensors or apparatus affixed to the subjects’ bodies. The implementation of this data collection method that minimizes interference guarantees a more authentic and unrestrained evaluation of gait patterns, thereby enhancing the ecological validity of the system.The system employs PoseNet, a deep learning model, to extract relevant features from videos that capture movements of the lower limbs. By utilizing the features of PoseNet, the system capitalizes on the model’s capacity to accurately estimate the human pose, facilitating a thorough examination of gait patterns.Through the application of machine learning (ML) algorithms to the extracted PoseNet features, the system possesses the capability to effectively classify and distinguish various disorders that impact the hip, ankle, and knee. The implementation of automation in this context serves to decrease the level of subjectivity involved in manual analysis, while also reducing the amount of time required for such analysis. As a result, the process of diagnosis becomes more expedient and efficient.

The aforementioned contributions play a significant role in the advancement of lower limb disorder classification, thereby holding the potential to yield substantial benefits for both clinical practice and research endeavors.

The subsequent sections of the article are structured as follows: Section 2 provides a comprehensive literature review, Section 3 outlines the methodology and experimental procedures, and the obtained results are discussed in Section 4. Finally, the conclusions are presented in Section 5.

## 2. Literature Review

In recent years, there has been growing interest in the field of gait analysis and the categorization of joint abnormalities. This attention is driven by the desire to enhance the accuracy of diagnosis and treatment methods. Various studies employ ML and deep learning techniques to automatically categorize joint abnormalities using gait data. Each study in the field of gait analysis is centered around a particular condition or aspect and employs a range of methodologies and evaluation metrics. The research conducted by [35] centers on the diagnosis of knee osteoarthritis using the automated analysis of walking data obtained from both diagnosed persons and symptom-free controls. Ground reaction force features are extracted using force plates and piezoelectric sensors, and these values are then associated with the severity of osteoarthritis using random forest regression models. The attained accuracy of 72.61% in the five-fold cross-validation indicates a decent level of performance, leaving space for potential improvement.

In [36], the researchers use supervised classifiers and an RGB-D camera to diagnose gait problems in osteoarthritis patients. The researchers categorize gait disorders with 97% accuracy using fourteen gait measures, demonstrating its potential for osteoarthritis diagnosis. Another work [37] proposes a novel method of detecting gait abnormalities using a single 2D video camera. Video analysis with a support vector machine (SVM) classifier determines biomechanical gait parameters with 98.8% accuracy. The research in [38] presents a cost-effective and user-friendly gait data acquisition and analysis system. This technique quantifies osteoarthritis-related walking irregularities. The hybrid prediction model, combining manual and automated characteristics, achieves 98.77% accuracy. Meanwhile, Ref. [39] uses deep learning to classify abnormal gait patterns by integrating 3D skeletal data and plantar foot pressure readings. The multimodal hybrid model achieves 97.60% accuracy by utilizing pressure and skeletal data effectively.

The aim of [40] was to develop an automated framework for knee osteoarthritis (KOA) classification utilizing radiographic imaging and gait analysis, with a Kallgren-Lawrence grading system. A support vector machine and deep learning features from Inception-ResNet-v2 classified KOA based on gait and radiographic data, showing strong relationships between gait characteristics and radiological severity. The AUC varied from 0.93 to 0.97 for KL grades 0–4. Moreover, Ref. [41] intended to evaluate gait symmetry in unilateral ankle osteoarthrosis (AOA) patients and identify variables affecting post-surgery asymmetry. They compared 46 gait metrics in 10 healthy people with 10 AOA patients using 3D inertial sensors and pressure insoles. They found significant differences in 23 impacted-side and 20 non-impacted-side variables. In particular, 14 metrics exhibited differences during bilateral AOA patient comparisons, notably in the toe area, and in forefoot mobility during walking.

In [42], the researchers use ground reaction force (GRF) measurements to automate the diagnosis of functional gait disorders (GDs). They evaluate GRF parameterization methods for GD identification and establish a reference for automatic classification. The study divides 279 GD patients and 161 healthy controls into hip, knee, ankle, and calcaneus impairment groups using GRF data. It tests GRF and PCA-based parameterization approaches. The evaluation of discriminative power uses linear discriminant analysis. The study classifies normal walking patterns and multiclass GD categories. The study in [43] focuses on categorizing gait disorders, with an emphasis on ground reaction force (GRF) analysis. The study preprocesses GRF signals and extracts and selects features from the GaitRec and Gutenberg databases with data from gait problem patients and healthy participants. The K-nearest neighbor (KNN) model outperforms conventional machine learning approaches in four experimental schemes categorizing gait disorders. The study contrasts vertical and three-dimensional GRF, with the latter performing better. Meanwhile, Ref. [44] develops an automated, accurate knee osteoarthritis (KOA) diagnosis method. The study uses RQA, fuzzy entropy, and statistical analysis to analyze dynamical characteristics collected from gait kinematic data. Discriminant analysis on these characteristics evaluates shallow classifiers like SVM, KNN, NB, DT, and Adaboost. SVM distinguishes KOA patients and healthy controls with the maximum accuracy of 92.31% and 100%, proving its KOA diagnostic efficacy.

Previous research studies provide evidence of the efficacy of ML and deep learning methodologies in the automated categorization of joint abnormalities using gait data. The utilization of diverse modalities, including RGB-D cameras, 2D video, and ground reaction force measurements, exemplifies the multifaceted nature of these methodologies. Nevertheless, certain studies demonstrate limitations in terms of moderate accuracy, the necessity for supplementary evaluation metrics, and comparatively limited sample sizes. This manuscript introduces a novel approach to categorizing lower limb disorders, focusing on ankle, knee, hip, and normal subjects. The proposed method involves the utilization of PoseNet features extracted from video data. The approach centers on utilizing PoseNet, a pose estimation model based on deep learning, to extract significant features from the video recordings. The primary objective of the proposed methodology is to improve the precision and effectiveness of diagnosing lower limb disorders through the utilization of video data. This approach capitalizes on the abundance of valuable information pertaining to subjects’ movements and joint positions that can be extracted from video recordings. The application of this methodology holds promise in assisting healthcare practitioners in the identification and classification of distinct lower limb disorders, thus facilitating the implementation of suitable treatment and rehabilitation approaches.

## 3. Materials and Methods

The methodology of this study was implemented on a Core i7 11th-generation machine running the Windows operating system. The proposed methodology was implemented using Jupyter Notebook [45], a web-based interactive tool that integrates live code execution, visualizations, and explanatory text. The Python programming language was used for the implementation. Several libraries, including sci-kit-learn and pandas, were utilized during the implementation process. Scikit-learn (sklearn) is a popular open-source machine learning library for Python that provides a comprehensive toolkit for classification, regression, and clustering [46]. Pandas is a well-known Python library for high-performance data analysis, with intuitive structures such as Series and DataFrame for structured data tasks such as cleansing, transformation, and aggregation [47].

### 3.1. Proposed Methodology

The research methodology employed for the classification of lower limb disorders is depicted in Figure 1. The data for this study were gathered at the Tehsil Headquarter (THQ) Hospital in Sadiq Abad through the application of a video camera. Then, the dataset was divided into training and testing sets. The frames were extracted from the videos to capture discrete temporal instances that were pertinent to the movement of the lower limbs. Subsequently, relevant features were derived from every frame utilizing the PoseNet algorithm. After performing feature extraction, the obtained features underwent standardization using a technique known as feature scaling. The standardized features were employed as inputs for a variety of ML algorithms. The algorithms underwent training and testing processes using a dataset to assess their efficacy in classifying various lower limb disorders, with a specific focus on hip, knee, and ankle ailments.

### 3.2. Data Collection

The research was carried out in compliance with ethical protocols and received approval from the ethics committee at Khwaja Fareed University of Engineering and Technology (KFUEIT). The committee conducted a thorough evaluation of the ethical implications of the study, guaranteeing adherence to the principles delineated in the Helsinki Declaration. Through the acquisition of ethical approval, the study effectively showcased its dedication to safeguarding the well-being, entitlements, and confidentiality of the individuals engaged in the research. The data collection was conducted at the THQ Hospital located in Sadiq Abad. The data were obtained from a group of 174 individuals, ranging in age from 40 to 60 years, encompassing both male and female participants. Figure 2 presents information pertaining to the number of participants and their corresponding genders. During the experiment, the participants were instructed to walk on a pathway measuring 9 m in length and data were recorded using a camera, as illustrated in Figure 3. The Hiievpu 2K Webcam was used for data collection as it exhibits sophisticated functionalities, effectively using the CMOS 1/s image sensor’s capability to provide high-definition visuals at a resolution of 4 million pixels (2560 × 1440p) [48]. This produces videos that are incredibly clear and operate smoothly at a frame rate of 30 frames per second [48]. To optimize patient comfort and convenience, all participants were given instructions to engage in a series of 10 walking sessions, with designated intervals of rest interspersed throughout. The objective of this approach was to acquire reliable data for subsequent processing, which involved extracting features and conducting further analysis using a video camera.

### 3.3. Feature Extraction

To achieve an accurate depiction of the lower limbs’ temporal features, frames were extracted from the videos at a rate of 30 frames per second (fps). The selection of this higher frame rate was made to record the motions linked to lower limb disorders precisely. Subsequently, the frames underwent the application of the PoseNet feature extraction technique. The PoseNet algorithm is extensively employed in the estimation of the human body pose [49]. The system examines the bodily posture in every frame and identifies significant landmarks, including joint positions, angles, and body key points. These landmarks offer valuable insights into the kinetics and spatial organization of the human body during walking. This study employed the Google MediaPipe library [50]; MediaPipe is a Google-developed open-source library that provides tools for the creation of perception pipelines that can process and analyze various forms of media data, such as images and videos, to extract information such as facial landmarks, hand tracking, and pose estimation. Here, the MediaPipe library was used to extract pose key points [28]. The MediaPipe library extracts 33 landmarks. In this research, only 22 distinct anatomical landmarks on the human body were extracted. The selection of 22 landmarks from the total of 33 was due to the exclusion of 11 head landmarks. The chosen landmarks represented the upper and lower limbs, as these segments are actively engaged during gait. The decision to include upper limb segment landmarks was influenced by the recognition of their roles beyond walking mechanics [51,52]. The movement of the arms and hands not only serves as a functional component of gait but also plays an essential role in maintaining balance [53,54,55]. During movement, the dynamic interplay between both the upper and lower extremities contributes to the overall stability and coordination of the human body. The landmarks extracted using the MediaPipe library are as follows.

Upper Body:-Left Shoulder;-Right Shoulder;-Left Elbow;-Right Elbow;-Left Wrist;-Right Wrist;-Left Pinky;-Right Pinky;-Left Index;-Right Index;-Left Thumb;-Right Thumb.Lower Body:-Left Hip;-Right Hip;-Left Knee;-Right Knee;-Left Ankle;-Right Ankle;-Left Heel;-Right Heel;-Left Foot Index;-Right Foot Index.

The mentioned landmarks were utilized as points of reference to denote the positioning of individuals within the frames of the video. The extraction of pose landmarks yielded significant data pertaining to the spatial positioning and arrangement of different body parts, such as the shoulders, elbows, wrists, hips, knees, and ankles. The accurate identification and analysis of joints and key points obtained through pose estimation play a vital role in effectively characterizing the posture and movements exhibited by individuals while walking. A widely employed method of representing pose landmarks involves the utilization of a skeletal representation. This process entails establishing connections between the pose landmarks through the use of lines, resulting in the formation of a skeletal framework that visually portrays the physical structure of the person, as illustrated in Figure 4. Only blue landmarks were selected in this study. Through the integration of key characteristics, a concise yet informative anatomical framework was established, showcasing the overall physical posture and organization. The X, Y, and Z coordinates of these pivotal points, along with their corresponding visibility scores, were derived from the Cartesian coordinate system [28]. In order to maintain the integrity of the data, they were subsequently stored in a Comma-Separated Values (CSV) file. The structure of the CSV file was such that each row corresponded to an individual instance or frame of the video. The columns within the file represented the X, Y, and Z coordinates of the pose landmarks, along with their respective visibility scores. This format guaranteed the systematic arrangement and convenient accessibility of the crucial data about the 22 pose landmarks, facilitating their subsequent analysis and processing.

Number of key points = 22.Number of extracted features from each key point = 4.Total number of features = 22 × 4 = 88.

### 3.4. Data Scaling and Feature Reduction

Feature scaling plays a pivotal role in the data preprocessing phase that follows feature extraction. The primary objective of this process is to ensure that all extracted features undergo a uniform transformation in relation to their scale or range. This step holds significance due to the potential for significant variances in value ranges among different features, which can impact the performance of specific ML algorithms. Feature scaling is a technique that mitigates the disproportionate influence of features with larger magnitudes on the model training process. The standard scaler technique was utilized in this manuscript for feature scaling [56]. The standard scaler is a widely employed technique for the normalization of feature values. The process involves centering the feature values by subtracting the mean and scaling them by dividing them by the standard deviation, resulting in a mean of zero and a standard deviation of one. The standard scaler accomplishes this by performing two operations on each data point: subtracting the mean value of the feature and dividing the result by the standard deviation. The equation for the standard scaler can be defined as
(1)$$X_{\mathrm{scaled}}=\frac{X-\mu }{\sigma }$$
where

$X_{\mathrm{scaled}}$ represents the scaled feature value;*X* is the original feature value;μ is the mean of the feature values in the dataset;σ is the standard deviation of the feature values in the dataset.

The normalization process guarantees that the feature values are standardized, irrespective of their initial distribution, by centering them around zero and applying uniform scaling. Following the application of the standard scaler to the dataset, the subsequent step in the methodology entailed the implementation of Principal Component Analysis (PCA) on the scaled features. PCA is a statistical methodology employed to achieve dimensionality reduction [57]. The main aim of this approach is to identify the most prominent patterns or variations within the dataset by converting the initial features into a distinct set of independent variables known as principal components. The dataset was represented by a selection of 80 principal components in this study. The selection of these components was based on their capacity to capture the greatest proportion of variance in the data while minimizing the loss of information. The principal components obtained were utilized for the training of different ML models.

### 3.5. Exploratory Data Analysis

The dataset consists of a heterogeneous compilation of 1740 videos, encompassing a range of categories including hip, ankle, knee, and normal. Figure 5, through informative pie charts, provides a visual representation of the distribution of videos in the training and testing sets among the different categories. The charts not only display the number of videos within each category but also emphasize the relative significance of each category by presenting their percentage representation in the overall dataset.

Upon analyzing Figure 5a,b, it becomes evident that there is a discernible pattern in the composition of the dataset. Specifically, the normal category displays a substantially greater number of contributions in comparison to the knee, ankle, and hip categories. The distribution of categories within the dataset offers valuable insights into their prevalence, enabling a comprehensive understanding of the dataset’s composition and its implications for both research and practical applications. The number of frames extracted from each category is shown in Figure 6.

The data distribution is represented by a cubic hyperplot in Figure 7. Figure 7a depicts a three-dimensional hyper-scatter plot of the dataset, facilitating a visual analysis of its distribution across three dimensions. In contrast, Figure 7b depicts a two-dimensional projection of the hyperplot, providing valuable observations regarding the linear separability of the data. The purpose of Figure 7 is to provide a comprehensive understanding of the data’s arrangement. The visualization facilitates the evaluation of how the data points are dispersed and whether or not they exhibit distinct patterns or separability. By analyzing Figure 7b, we can see that the data points exhibit a distinct linear separation. The choice between a three-dimensional and a two-dimensional representation enables the development of distinct insights. The three-dimensional perspective provides a holistic view of the data distribution, whereas the two-dimensional projection emphasizes the linear relationships within the data. Understanding the dimensionality and separability of the data is essential in selecting suitable classification methods and determining the potential accuracy of linear classifiers in categorizing the data points. This observation suggests that the data can be successfully partitioned into discrete categories using a linear classifier. The presence of linear separability in the dataset implies that it is possible to utilize an easy decision boundary to effectively classify the data points.

## 4. Results and Discussion

The dataset was split into a training set and a testing set using a ratio of 70% and 30%, respectively, in order to compare the performance of several ML models. This partitioning facilitated the evaluation of the model’s performance on data that had not been previously observed. We used various widely used ML algorithms, including Random Forest (RF), Extra Tree Classifier (ETC), K-nearest neighbor (KNN), Adaboost, and Multilayer Perceptron (MLP), as well as deep learning models like Artificial Neural Networks (ANN) and Convolutional Neural Networks (CNN). These ML models are used in different real-time applications related to disease diagnosis [58], computer vision [59,60], agriculture [61], and education [62], among others. Grid Search was used for hyperparameter optimization to enhance the performance of the models. The goal of Grid Search is to find the highest possible model performance by systematically searching through different hyperparameter combinations. The hyperparameters used in this study are shown in Table 1. The numbers 1024, 512, 256, and 128 in Table 1 correspond to the specific configurations of hidden layers and neuron counts within the ANN architecture in our study. These values indicate the number of neurons present in each hidden layer of the ANN.

K-fold cross-validation was performed to assess the generalizability of the trained models. K-fold cross-validation is a technique that divides a dataset into K subgroups and trains and evaluates a model K times, once for each subset as a validation set, to analyze and improve the model’s performance and generalization [63]. In this study, the dataset was separated into five folds, making sure that each fold represented an equal and representative percentage of the data. The utilization of this methodology facilitated a thorough evaluation of the models’ efficacy across various partitions of the dataset, yielding valuable insights into their ability to adapt and generalize to unfamiliar data. After performing K-fold validation, the models were assessed using the designated testing set. The frames were extracted from the videos in the testing set, and a set of preprocessing and feature engineering techniques were employed to improve the representation of the data. Subsequently, the classifiers that had undergone training proceeded to generate predictions for every individual frame within the video. The predictions obtained from each frame were consolidated, and the prediction that occurred most frequently was selected as the ultimate prediction for the entire video. The proposed methodology takes into consideration temporal data and effectively captures the overarching pattern present in the video, thereby enhancing the dependability and precision of the prediction. Through the utilization of this approach of frame-level prediction and aggregation, the models can proficiently classify videos by analyzing the content and patterns present in multiple frames. The methodology employed in this study utilizes ML algorithms to effectively analyze and interpret video data, thereby yielding significant insights into the identification and assessment of particular disorders or conditions. The classification matrix along with the K-fold scores are shown in Table 2.

The data presented in Table 2 indicate that the MLP, ANN, and CNN models exhibited the most notable levels of accuracy, which varied between 97.88% and 98.84%. The RF, Adaboost, KNN, and ETC models demonstrated accuracy of approximately 94%, accompanied by elevated precision, recall, and F1-Score metrics. The MLP achieved an accuracy rate of 97.88% along with exceptional precision and recall metrics. The ANN and CNN demonstrated exceptional performance, achieving a remarkable accuracy rate of 98.84%. Furthermore, these models exhibited near-flawless precision, recall, and F1-Score values. Nevertheless, there was observed variability in the cross-validation scores for the ANN and CNN models. The ANN achieved a better validation score than the CNN. The confusion matrix and accuracy loss curve of the ANN are shown in Figure 8.

The confusion matrix presented in Figure 8a depicts the classification outcomes of a predictive model focused on lower limb disorders, specifically targeting ankle, hip, knee, and normal conditions. The model demonstrated robust performance across all categories, accurately predicting the majority of instances within each class. The classification of the ankle class (class 0) yielded accurate predictions, with 105 instances out of 105 correctly classified. The hip class (class 1) achieved a total of 65 accurate predictions out of 67, with a single misclassification each in knee and ankle. In the knee class (class 2), a total of 174 instances out of 178 were accurately classified. However, three were wrongly classified as ankle and one as hip, while these instances actually belonged to knee. All 169 instances in the normal class (class 3) were accurately classified. Although the model exhibited a commendable level of accuracy, it did encounter a minute number of misclassifications, specifically in distinguishing between the hip and knee classes. The data presented in Figure 8b demonstrate a decreasing trend in loss over epochs, indicating that the model’s predictive performance improves over time. Additionally, there is a corresponding increase in accuracy, suggesting that the model becomes more proficient in making accurate predictions as it learns. The loss consistently decreases and eventually reaches an equilibrium point, which suggests that the model is exhibiting convergence and effectively acquiring knowledge from the provided data. The class-wise classification matrix of the ANN is given in Table 3.

Table 3 shows that the ANN model performed well across all classes. The achieved precision of 0.96 for the “ankle” class signifies that when the ANN identifies an instance as belonging to the “ankle” class, it is accurate 96% of the time. In other words, the ANN has a strong level of correctness in correctly classifying instances as “ankle”. A recall value of 1.00 indicates that the model detected all occurrences of the “ankle” class among the true positive instances. The F1-Score of 0.98 signifies a favorable equilibrium between precision and recall. In the case of the “hip” class, the ANN model demonstrated a precision score of 1.00, accurately classifying all instances as members of the “hip” class. The recall value of 0.97 indicates that the model effectively detected 97% of the true positive instances for the “hip” class. The F1-Score, with a value of 0.98, indicates well-balanced performance in terms of both precision and recall. The ANN model demonstrated a precision value of 0.99 for the “knee” class, suggesting precise predictions for instances categorized under the “knee” class. The recall value of 0.98 indicates that the model effectively detected 98% of the true positive instances pertaining to this particular class. The F1-Score, which is calculated as the harmonic mean of precision and recall, exhibits a noteworthy equilibrium between these two performance metrics, with a value of 0.99. Finally, the “normal” class demonstrated exceptional outcomes, with a precision and recall score of 0.99, signifying precise predictions and the accurate identification of all positive instances. The F1-Score of 1.00 demonstrates an optimal equilibrium between precision and recall for this particular class.

### 4.1. Computational Complexity

This study examined the computational time complexities associated with classifiers used for the classification of lower limb disorders using video data. The analysis of computational complexity was conducted with a focus on the hyperparameters that resulted in higher accuracy. The experiments were performed using an HP ProBook 450 G4 laptop equipped with 16 GB of RAM and an Intel Core i5 7th generation processor. The findings in Table 4 indicate that the ETC classifier exhibited the lowest computational time complexity, with a duration of 102 s. This was followed by the MLP classifier, which took 128 s; KNN, which took 155 s; and the RF classifier, which took 364 s. The Adaboost and ANN classifiers exhibited computational time complexity of 462 and 500 s, respectively, whereas the CNN demonstrated the highest complexity of 712 s.

The MLP stands out as a strong competitor in light of the performance metrics and computational time complexity. The MLP demonstrates superior performance in terms of accuracy, precision, recall, and F1-score while exhibiting a comparatively lower level of computational time complexity. The ANN and CNN also exhibit exceptional performance; however, their computational time complexities are comparatively higher than those of the MLP.

### 4.2. Comparison with Existing Studies

The primary objective of the proposed study was to categorize lower limb disorders by utilizing PoseNet features that were extracted from video data. The findings of the study demonstrate a high level of encouragement, as evidenced by the attained accuracy rate of 98.8% and the cross-validation score of 99%. These results highlight the potential of the proposed method in achieving precise classification. Upon comparing the current study with previous research conducted on knee osteoarthritis [35] and gait abnormalities [36], it becomes evident that the current study shows notable accuracy. Furthermore, the accuracy of the current method is similar to that of studies that employed 2D video camera data [38] and cost-effective systems for the analysis of gait. However, the primary objective of the proposed study focused on a wider range of lower limb disorders, encompassing not only gait abnormalities associated with osteoarthritis but also other conditions. Furthermore, the present study showcases comparable performance to previous studies that employed 3D skeletal data and foot pressure measurements [39]. Moreover, it exhibits favorable comparisons to studies that utilized radiographic imaging and gait analysis data, which reported area under the curve (AUC) values ranging from 0.82 to 0.97 [18]. In general, the study presents encouraging results, indicating the potential of utilizing PoseNet features extracted from video data as a viable method in classifying lower limb disorders. In contrast to prior research endeavors that depended on expensive and intricate systems, such as 3D skeletal data and foot pressure measurements [39], the present study employed video data and PoseNet features for analysis. The accessibility, cost-effectiveness, and user friendliness of video data render it a practical option for widespread implementation and potential utilization in clinical settings. The comparison of the studies is given in Table 5.

## 5. Conclusions

This research introduces a novel method of classifying lower limb problems based on gait analysis and PoseNet features, with an emphasis on the knee, hip, and ankle. To obtain detailed information about the bio-mechanics of the lower limb, PoseNet is used to extract important body joint movements and positions from video footage in a non-invasive manner. After feature extraction and feature engineering, several machine and deep learning models were trained and tested on the dataset. The results show that the suggested method is highly accurate and precise in the classification of lower limb diseases, with accuracies ranging from 93.44% to 98.84%. Non-invasiveness, user friendliness, and the ability to capture natural gait patterns are just a few of the benefits of this method. It has the potential to help medical personnel to effectively identify and precisely diagnose lower limb disorders, allowing for targeted treatment and rehabilitation techniques. Several directions could be taken by researchers in the future. First, improving the models’ applicability would require an increase in the size and diversity of the dataset. The validity and clinical utility of this technique could be further confirmed by exploring its application to various illnesses and conditions affecting the lower limbs.

## Figures and Tables

**Figure 1 diagnostics-13-02881-f001:**
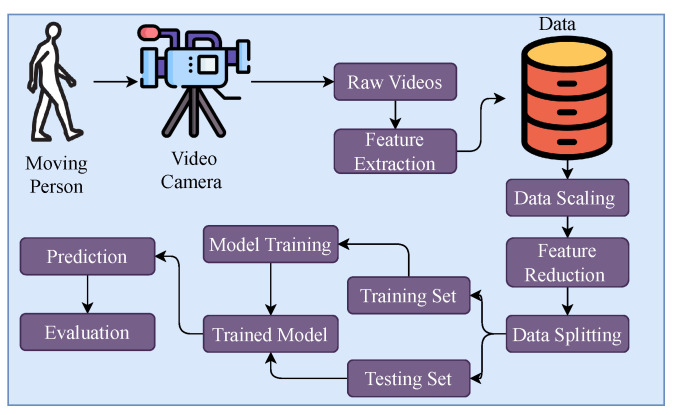
Proposed methodology diagram.

**Figure 2 diagnostics-13-02881-f002:**
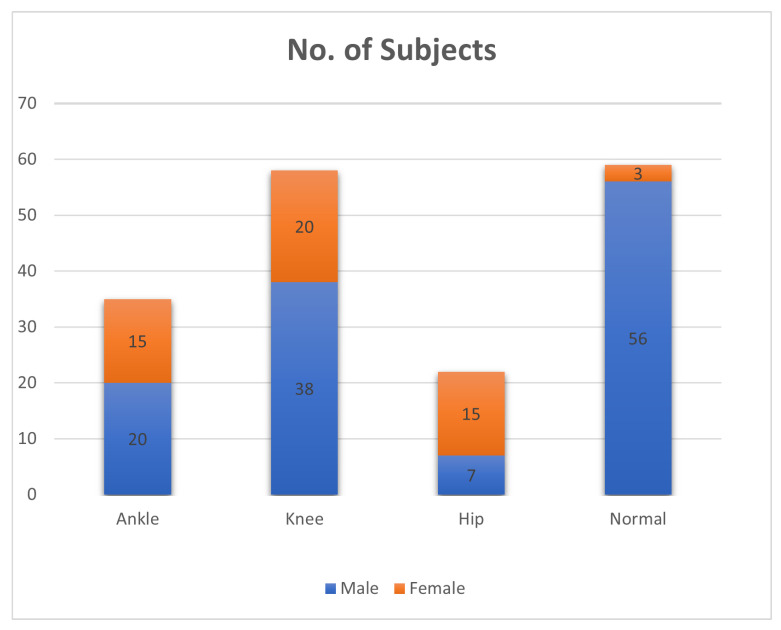
Distribution of subjects and genders.

**Figure 3 diagnostics-13-02881-f003:**
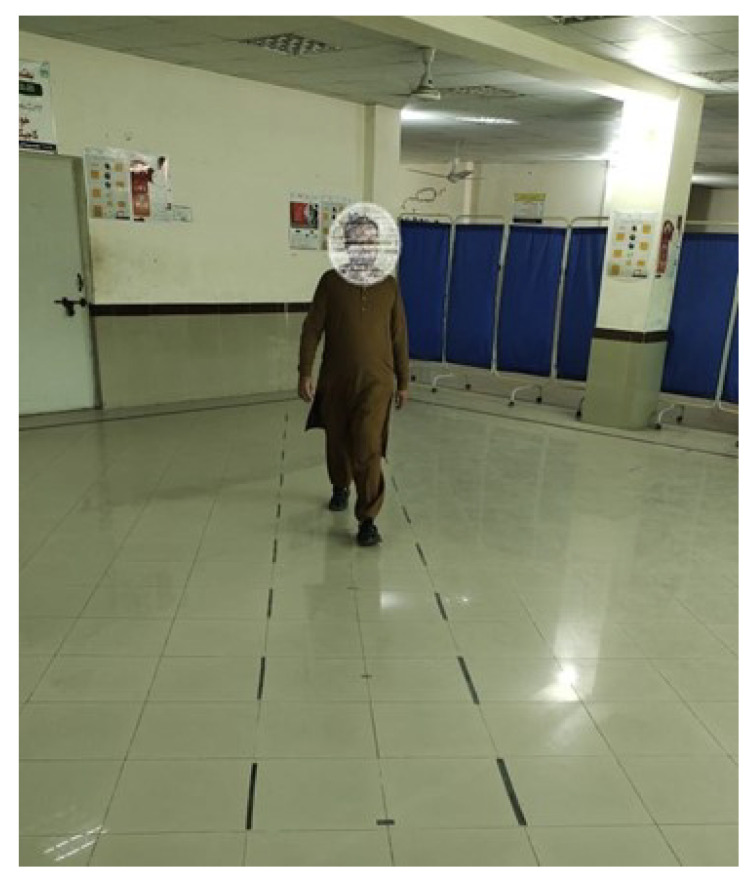
Subject walking on the walkway in front of the camera.

**Figure 4 diagnostics-13-02881-f004:**
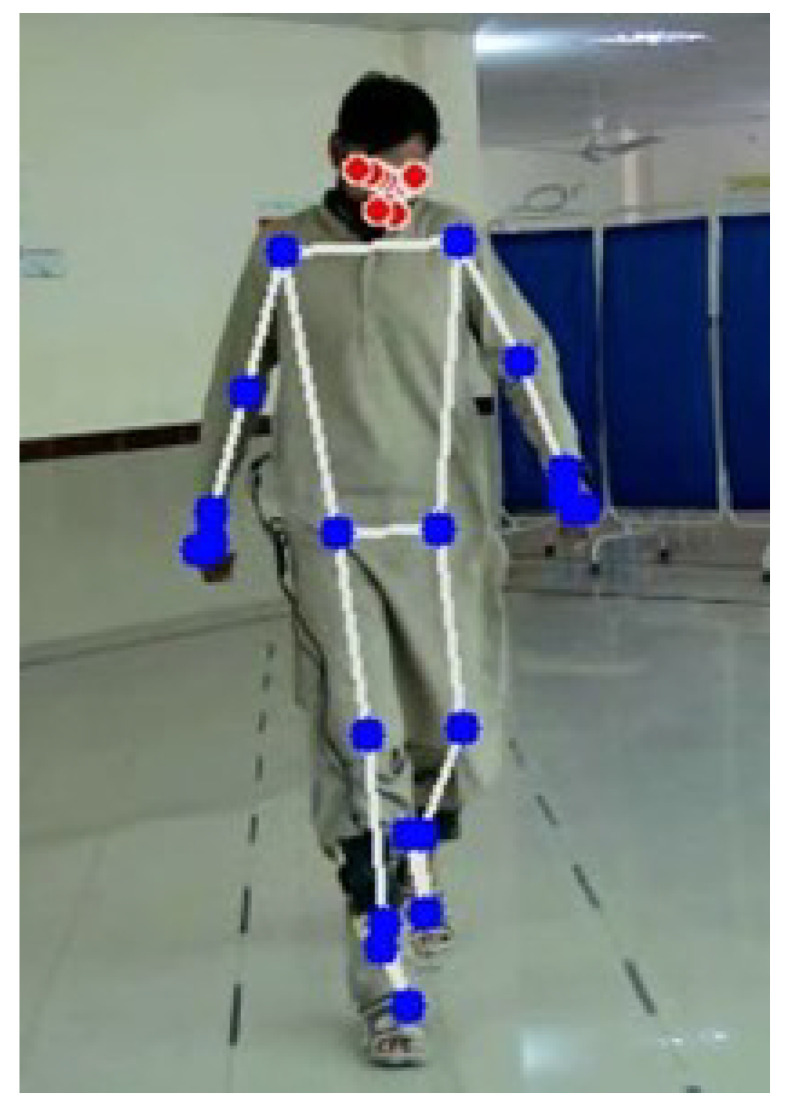
Selected landmarks of subject in blue.

**Figure 5 diagnostics-13-02881-f005:**
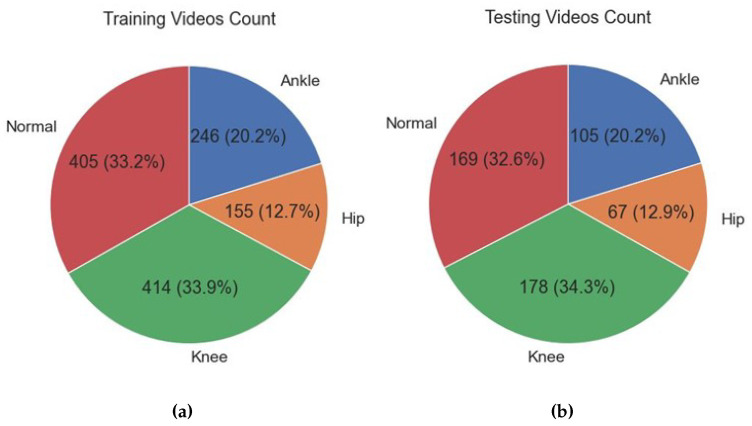
Distribution of videos in each category: (**a**) training set (**b**) testing set.

**Figure 6 diagnostics-13-02881-f006:**
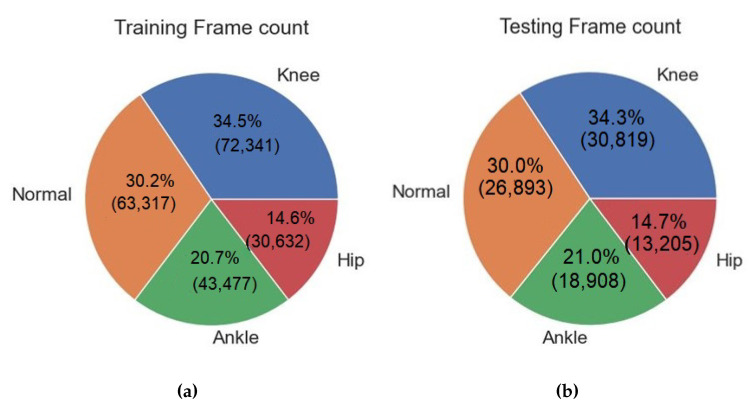
Distribution of frames in different categories: (**a**) training set (**b**) testing set.

**Figure 7 diagnostics-13-02881-f007:**
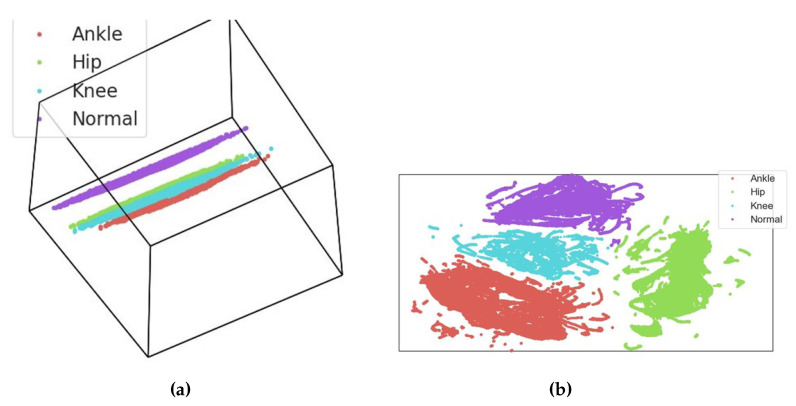
(**a**) Cubic hyperplot of the dataset. (**b**) A 2D projection of the hyperplot.

**Figure 8 diagnostics-13-02881-f008:**
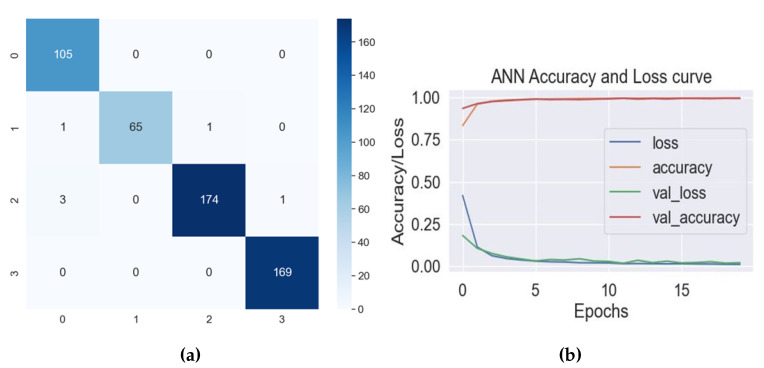
(**a**) Confusion matrix of ANN. (**b**) Accuracy loss curve of ANN.

**Table 1 diagnostics-13-02881-t001:** Hyperparameters used to tune the classifiers.

Classifier	Hyperparameters
RF	random_state=100, max_depth=50, n_estimators=100
ETC	n_estimators=100, max_depth=200, random_state=0
Adaboost	ExtraTreesClassifier(n_estimators=100, max_depth=200, random_state=0)
KNN	algorithm=‘auto’, leaf_size=50, metric=‘minkowski’, metric_params=None, n_jobs=3, n_neighbors=4, weights=‘uniform’
MLP	random_state=142, max_iter=100
ANN	Dense (1024, activation=‘relu’), Dense (512, activation=‘relu’), Dense (256, activation=‘relu’), Dense (128, activation=‘relu’), optimizer=‘adam’, loss=‘categorical_crossentropy’, metrics=[‘accuracy’]
CNN	Conv1D (32, 3, activation=‘relu’), MaxPooling1D (2), Conv1D (64, 3, activation=‘relu’) MaxPooling1D (2), Flatten (), Dense (128, activation=‘relu’), Dense (num_classes, activation=‘softmax’), optimizer=‘adam’, loss=‘categorical_crossentropy’, metrics=[‘accuracy’]

**Table 2 diagnostics-13-02881-t002:** Classification matrix and K-fold scores of the classifiers.

Classifier	Accuracy (%)	Precision	Recall	F1-Score	K-Fold Cross- Validation Score
RF	94	0.96	0.91	0.93	0.97 ± 0.00
ETC	93.44	0.96	0.91	0.93	0.98 ± 0.00
KNN	95.76	0.96	0.95	0.96	0.98 ± 0.00
Adaboost	93.06	0.95	0.90	0.92	0.98 ± 0.00
MLP	97.88	0.98	0.98	0.98	0.95 ± 0.00
ANN	98.84	0.99	0.99	0.99	0.99 ± 0.02
CNN	98.84	0.99	0.99	0.99	0.97 ± 0.10

**Table 3 diagnostics-13-02881-t003:** Class-wise classification accuracy of ANN.

Class	Precision	Recall	F1-Score
Ankle	0.96	1.00	0.98
Hip	1.00	0.97	0.98
Knee	0.99	0.98	0.99
Normal	0.99	1.00	1.00

**Table 4 diagnostics-13-02881-t004:** Computational time complexity of classifiers.

Classifier	Computational Time Complexity (s)
RF	364
ETC	102
KNN	155
Adaboost	462
MLP	128
ANN	500
CNN	712

**Table 5 diagnostics-13-02881-t005:** Comparison with existing studies on lower limb disorder classification.

Study Reference	Focus	Accuracy/Results
[35]	Automated detection of knee osteoarthritis	Mean accuracy: 72.61%
[36]	Assessment and diagnosis of gait abnormalities in osteoarthritis	Average accuracy: 97%
[37]	Automated detection and classification of gait abnormalities using a 2D video camera	Accuracy: 98.8%
[38]	Cost-effective system for acquiring and analyzing gait data in osteoarthritis	Accuracy: 98.77%
[39]	Classification of atypical gait patterns using 3D skeletal data and foot pressure measurements	Accuracy: 97.60%
[40]	Automated categorization framework for knee osteoarthritis using radiographic imaging and gait analysis data	AUC values range from 0.82 to 0.97
[44]	Diagnostic system for knee osteoarthritis using dynamical gait features	SVM classifier accuracy: 92.31% (KOAs vs. controls), 100% (healthy controls)
Proposed Study	Classification of lower limb disorders using PoseNet features extracted from video data	ANN accuracy: 98.8%, CV score: 99% (std. dev.: 0.02)

## Data Availability

Data will be provided on demand.

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
