# Peer review of "Empowering Lower Limb Disorder Identification through PoseNet and Artificial Intelligence"

_diagnostics, 2023, doi:10.3390/diagnostics13182881_

Round 1

Reviewer 1 Report

Introduction:

1.     References should be added after the sentence of “These conditions commonly resulting from trauma, degenerative diseases, or bio-mechanical 23 abnormalities, can give rise to discomfort, restricted range of motion, and diminished functionality”.

2.     References should be added after each of the “domains” in the sentence of “Gait analysis is employed in a wide variety of domains, including medical diagnostics, osteopathic medicine, comparative bio-mechanics, and sports-related bio-mechanics.”

3.     References should be added after the sentence of “Through the examination of these gait parameters, medical professionals and scholars can detect irregularities or deviations from typical gait patterns, which may serve as indicators of the existence of a lower limb disorder.”

4.     What is PoseNet? Is it a software or algorithm? If it is an existing product, please use in-text citation. 

5.     If PoseNet has been used in any other studies? Please add references here. Why this study chose PoseNet?

6.     I don’t think that the first bullet point “The data was collected from a total of 174 real patients and normal individuals …” belongs to contribution. It should be included into the method section.

Literature:

1.     It is not necessary to have two and half pages for literature review. You can have one or two paragraphs for the literature review, but the most important is the limitations of the previous studies, and the novelty of your study.

2.     Please explain what is Jupyter Notebook, and what is sci-kit-learn and pandas.

Methods:

1.     What cameras did you use in this study? Please explain.

2.     Please explain “Google Media Pipe Library”.

3.     Why you chose 22 landmarks from the 33 landmarks? What is the principle?

4.     If you want to analyze the lower limb disorders, why do you need the landmarks of pinky, index and thumb? I don’t think the movement of fingers can help to predict lower limb disorders. Please explain.

5.     For the skeletal framework, it seems that you used a different number of landmarks. Please explain.

6.     Why you chose 4 as the number of extracted from each key point?

7.     Figure 5 and figure 6 are not clear. Please use better pixel pictures.

8.     What is the point of having figure 7? Why the 3-d or 2-d data distribution is important?

Results:

1.     Table 2 is not clear. For example, why you mentioned 1024, 512 and 215 for ANN?

2.     What is K-fold cross validation?

3.     From the results, you have the conclusion that ANN achieved a better validation score than CNN. Did you consider the structure of each model? If you add more layers or neurons in the CNN, probably the results are different. Please explain.

4.     Figure 8 is redundant, because you already have Table3.

5.     Figure 9 needs more explanation. Why do you have to mention it in this paper? Why is it important?

6.     Figure 10 is redundant, because you already have Table 4.

7.     For your computational complexity, the results totally depend on the structure of each model. Thus, you need more explanation to make the results to be comparable.

8.     Figure 11 is redundant.

Conclusion:

1.     It is difficult to see the novelty of this study. What is new in this study? It seems that this study just used an existing software to compare different ML models.

References:

1.     A technical paper should include approximately 40 references. Please add more into your manuscript. 

na

Author Response

Reviewer 1

Concern 1: References should be added after the sentence of “These conditions commonly resulting from trauma, degenerative diseases, or bio-mechanical abnormalities, can give rise to discomfort, restricted range of motion, and diminished functionality”.

Response: We would like to thank the reviewer for the valuable comments to improve the quality of the manuscript. We added references according to the reviewer's suggestion.

Concern 2: References should be added after each of the “domains” in the sentence of “Gait analysis is employed in a wide variety of domains, including medical diagnostics, osteopathic medicine, comparative bio-mechanics, and sports-related bio-mechanics.”

Response: We update the manuscript by adding references according to the reviewer suggestion.

Concern 3: References should be added after the sentence of “Through the examination of these gait parameters, medical professionals and scholars can detect irregularities or deviations from typical gait patterns, which may serve as indicators of the existence of a lower limb disorder.”

Response: We update the manuscript by adding references according to reviewer suggestion.

Concern 4: What is PoseNet? Is it software or algorithm? If it is an existing product, please use in-text citation.

Response: PoseNet is a real-time pose estimation model developed by Google, which employs neural network techniques to accurately assess human poses from both photos and videos [13].

Concern 5: If PoseNet has been used in any other studies? Please add references here. Why did this study choose PoseNet?

Response: Reviewer suggestion is valuable and we added references of other studies using PoseNet. PoseNet was chosen by the author because of its better attributes in the domain of real-time human pose estimation. This is due to its real-time capabilities, user-friendliness, and versatility. The model's ability to perform efficiently with just one video feed, together with its minimal preprocessing requirements and smooth integration into widely used deep learning frameworks, makes it an excellent choice for a wide range of applications.

Concern 6: I don’t think that the first bullet point “The data was collected from a total of 174 real patients and normal individuals …” belongs to contribution. It should be included into the method section

Response: In response to the reviewer's question, we would like to emphasize that one of our study's significant contributions is data collection. The team carefully gathered the data, and this work is recognized as a substantial contribution.

Concern 7: It is not necessary to have two and a half pages for the literature review. You can have one or two paragraphs for the literature review, but the most important are the limitations of the previous studies and the novelty of your study.

Response: Limitations of studies are discussed in the last paragraph of the literature review. Also, comparison of the existing study with previous studies is added.

Concern 8: Please explain what is Jupyter Notebook, and what is sci-kit-learn and pandas.

Response: The proposed methodology was implemented using Jupyter Notebook [ 23], a web-based interactive tool that integrates live code execution, visualization, and explanatory text. The Python programming language was used for the implementation. Scikit-learn (sklearn) is a popular open-source machine learning library for Python that provides a comprehensive toolkit for classification, regression, and clustering [24]. Pandas is a well-known Python library for high-performance data analysis, with intuitive structures such as Series and DataFrame for structured data tasks. such as cleansing, transformation, and aggregation [25].

Concern 9: What cameras did you use in this study? Please explain.

Response: The Hiievpu 2K Webcam is used that exhibits sophisticated functionalities, effectively using the CMOS â…“ image sensor's capability to provide high-definition visuals at a resolution of 4 million pixels (2560 x 1440p). This produces videos that are incredibly clear and operate smoothly at a frame rate of 30 frames per second.

Concern 10: Please explain “Google Media Pipe Library”.

Response: This study employed the Google Media Pipe library [28], MediaPipe is a Google-developed open-source library that provides tools for creating perception pipelines that can process and analyse various forms of media data, such as images and videos, to extract information such as facial landmarks, hand tracking, and pose estimation. Here in this manuscript the MediaPipe library is used to extract pose key point [ 29].

Concern 11: Why did you choose 22 landmarks from the 33 landmarks? What is the principle?

Response: The selection of 22 landmarks from the total of 33 was guided by the exclusion of 11 head landmarks. The chosen landmarks represent the upper and lower limbs, as these segments are actively engaged during gait. The decision to include upper limb segment landmarks was influenced by the recognition of their role beyond walking mechanics [50, 51]. The movement of the arms and hands not only functions as functional components of gait but also plays an essential role in maintaining balance [ 52 –54]. During movement, the dynamic interplay between both the upper and lower extremities contribute to the overall stability and coordination of the human body.

Concern 12: If you want to analyze lower limb disorders, why do you need the landmarks of the pinky, index and thumb? I don’t think the movement of fingers can help to predict lower limb disorders. Please explain.

Response: The authors selected these landmarks because the movement of the arms and hands not only functions as functional components of gait, but also plays an essential role in maintaining balance [ 52 –54].

Concern 13: For the skeletal framework, it seems that you used a different number of landmarks. Please explain.

Response: Only Blue landmarks were used in this study, and it is also explicitly mentioned in the caption of figure 4.

Concern 14: Why you chose 4 as the number of extracted from each key point?

Response: Every landmark includes its position, in x, y, and z coordinates, as well as a visibility indicator. This leads in a total of four distinctive features for each unique landmark.

Concern 15: Figure 5 and figure 6 are not clear. Please use better pixel pictures.

Response: We would like to thank the reviewer for the valuable suggestions. Figures have been updated.

Concern 16: What is the point of having figure 7? Why the 3-d or 2-d data distribution is important?

Response: The purpose of Figure 7 is to provide a comprehensive understanding of the data's arrangement. The visualization facilitates the evaluation of how the data points are dispersed and whether or not they exhibit distinct patterns or separability. By analyzing Figure 7 (b), we can see that the data points exhibit a distinct linear separation. The choice between a three-dimensional and a two-dimensional representation enables the development of distinct insights. The three-dimensional perspective provides a holistic view of data distribution, whereas the two-dimensional projection emphasizes the linear relationships within the data. Understanding the dimensionality and separability of the data is essential for selecting suitable classification methods and determining the potential accuracy of linear classifiers in categorizing the data points.

Concern 17: Table 2 is not clear. For example, why you mentioned 1024, 512 and 215 for ANN?

Response: The numbers 1024, 512, and 256 correspond to the specific configurations of hidden layers and neuron counts within the Artificial Neural Network (ANN) architecture in our study. These values indicate the number of neurons present in each hidden layer of the ANN.

Concern 18: What is K-fold cross validation?

Response: K-Fold Cross-Validation is a technique that divides a dataset into K subgroups and trains and evaluates a model K times, once for each subset as a validation set, to analyze and improve the model's performance and generalization.

Concern 19: From the results, you have the conclusion that ANN achieved a better validation score than CNN. Did you consider the structure of each model? If you add more layers or neurons in the CNN, probably the results are different. Please explain.

Response: Regarding your inquiry regarding the model architecture selection procedure, we would be happy to provide additional information. In our study, we exhaustively investigated a number of architectures to identify the one that would produce the highest accuracy score for the classification of lower limb disorders using video data. This exploratory phase involved meticulously evaluating various layer, neuron, and hyperparameter combinations. This exhaustive search and evaluation procedure resulted in the model architecture presented in the manuscript.

Concern 20: Figure 8 is redundant, because you already have Table3. Figure 10 is redundant, because you already have Table 4. Figure 11 is redundant.

Response: We appreciate the reviewer's thorough analysis of our work. In our study, the presence of interactive figures alongside the tables serves a specific function in improving the reader's understanding and engagement. While tables show information concisely, interactive figures give readers a dynamic visual representation that allows for deeper investigation and interpretation of the data. These graphs enable users to analyze trends, patterns, and specific data points interactively, which can lead to more nuanced findings. We aim to present our readers with a thorough and interactive experience that best supports their comprehension of the results by mixing tables and interactive figures.

Concern 21: Figure 9 needs more explanation. Why do you have to mention it in this paper? Why is it important?

Response: Figure 9 of our paper illustrates the confusion matrices of our proposed ANN model as well as the accuracy loss curve. This visual representation reveals the classification performance and training dynamics of the model. It assists the reader in comprehending the model's predictive capabilities, training progression, and potential accuracy fluctuations.

Concern 22: For your computational complexity, the results totally depend on the structure of each model. Thus, you need more explanation to make the results to be comparable.

Response: The computational complexity analysis took into consideration the impact of different model structures, with a focus on hyperparameters driving improved accuracy. This method ensures a fair comparison of results within the context of our research.

Concern 23: It is difficult to see the novelty of this study. What is new in this study? It seems that this study just used existing software to compare different ML models.

Response: Several distinct elements contribute to the originality of our investigation. First, we meticulously collected a custom dataset to ensure its applicability and specificity to the classification of lower limb disorders. This data acquisition procedure laid the groundwork for the originality of our research. Second, we utilized PoseNet features extracted from video data, a technique that has received less attention in the context of lower limb disorders. This selection of features adds a novel perspective to the analysis and classification of these conditions. In addition, our research went above and beyond typical model comparisons by exhaustively investigating various hyperparameter configurations. This emphasis on parameter adjusting acknowledges the importance of optimizing model settings for optimal outcomes.

Concern 24: A technical paper should include approximately 40 references. Please add more into your manuscript.

Response: References increased.

Reviewer 2 Report

-The paper presents an approach that uses gait analysis and PoseNet feature for the classification of lower limb disorders, in particular those related to knee, hip and ankle.

-Interesting research and proposed method that could be very helpful in medical real-world settings to support patients with lower limb disorders.

-A number of machine learning algorithms (Random Forest, Extra Tree Classifier, Multilayer Perceptron, Artificial Neural Networks, and Convolutional Neural Networks) were used for the classification with the Artificial Neural Networks model achieving the highest accuracy result of 98.84%.

-In Line 115, the start of the sentence reads a bit redundant: "The utilization of deep learning techniques is employed in...". It could be re-written as: "Deep learning techniques are employed in...".

-The distribution of participants categorised as "Normal" is very unbalanced, having 56 males and 3 females, I think the authors should have indicated the reason for this as well as considering how that would affect the generation of the model.

-What was the rationale of including in your research 22 and not all the 33 anatomical landmarks extracted by the Media Pipe library?

-The organisation and layout of Table 1 is a bit unusual, I would suggest to present the landmarks extracted in a more logical order in a table form or as a list in the main text.

-In Line 276 I think there is a missing word after "extracted".

-The equation for the standard scaler technique is missing after Line 292.

-The graphs shown in Figures 5, 6, 7, 9 and 11 are a bit blurry, I suggest to use better quality images.

-From the conclusion, it is not clear if the proposed method is already being used in a real-world medical setting or if it is mentioned in the context of the paper.

-In Line 115, the start of the sentence reads a bit redundant: "The utilization of deep learning techniques is employed in...". It could be re-written as: "Deep learning techniques are employed in...".

-In Line 276 I think there is a missing word after "extracted".

-Regarding the writing style and typos in the paper, I suggest that the authors proofread the paper as there are some typos.

Author Response

Reviewer 2

Concern 1: In Line 115, the start of the sentence reads a bit redundant: "The utilization of deep learning techniques is employed in...". It could be re-written as: "Deep learning techniques are employed in...".

Response: We would like to thank the reviewer for the valuable comments. The line is changed to “Deep learning techniques are employed in” in the manuscript.

Concern 2: The distribution of participants categorized as "Normal" is very unbalanced, having 56 males and 3 females, I think the authors should have indicated the reason for this as well as considering how that would affect the generation of the model.

Response: Due to cultural norms and sensitivities, obtaining consent and assuring balanced data collection, particularly from female participants, can be challenging in predominantly Muslim regions such as Pakistan. That yields the unbalancing of the dataset.

For both male and female patient data, the performance of the proposed ANN technique was evaluated independently. The results indicate that the model produced comparable outcomes for male and female patients, indicating the absence of a substantial gender bias. The exceptionally high accuracy, precision, recall, and F1 scores observed in both groups provide tangible evidence supporting the fairness and neutrality of the proposed method for managing gender-related data.

Parameters

Male

Females

Accuracy

97.43

97.9

Percision

97

98

Recall

97

98

F1-score

97

98

Concern 3: What was the rationale of including in your research 22 and not all the 33 anatomical landmarks extracted by the Media Pipe library?

Response: The selection of 22 landmarks from the total 33 was guided by the exclusion of 11 head landmarks. The chosen landmarks represent upper and lower limbs, as these segments are actively engaged during gait. The decision to include upper limb segment landmarks was influenced by the recognition of their role beyond walking mechanics [50, 51]. The movement of the arms and hands not only functions as functional components of gait but also plays an essential role in maintaining balance [ 52 –54]. During movement, the dynamic interplay between both the upper and lower extremities contributes to the overall stability and coordination of the human body.

Concern 4: The organization and layout of Table 1 is a bit unusual, I would suggest to present the landmarks extracted in a more logical order in a table form or as a list in the main text.

Response:  The landmarks are presented logically in the manuscript and changes are made.

Concern 5: In Line 276 I think there is a missing word after "extracted".

Response: The line has been corrected as “Number of extracted features from each key point”.

Concern 6: The equation for the standard scaler technique is missing after Line 292.

Response: We added Equation in updated manuscript according to the reviewer's suggestion.

Concern 7: The graphs shown in Figures 5, 6, 7, 9 and 11 are a bit blurry, I suggest to use better quality images.

Response: Figures have been improved.

Concern 8: From the conclusion, it is not clear if the proposed method is already being used in a real-world medical setting or if it is mentioned in the context of the paper.

Response: The reviewer concern is valuable and we update the conclusion and changes can be found in updated in the manuscript.

Reviewer 3 Report

The article presents a novel approach to the classification of lower limb disorders. The approach uses gait analysis and the extraction of PoseNet features from video data to effectively identify and categorize these disorders.

Overall, the article provides a comprehensive overview of a novel approach to the classification of lower limb disorders. The approach is based on gait analysis and the extraction of PoseNet features from video data. The results of the study demonstrate the potential of the approach for enhancing the diagnosis and treatment planning of lower limb disorders.

I recommend improving the quality of figures 5-7.

Author Response

Reviewer 3

Concern 1: I recommend improving the quality of figures 5-7.

Response: Reviewer's suggestion is valuable to improve the quality of the manuscript. Figures have been improved.

Reviewer 4 Report

Dear authors,

Thank you for the opportunity to review your paper. I have very few issues with the paper. I do ask for the following: ensure that the tables are formatted correctly as per the suggested MDPI style and that they are formatted the same throughout, and update the images so they are not blurry (maybe use Snippet app).

I also suggest collapsing the literature review into the introduction as the introduction section serves the purpose of providing a review of the literature. 

I have also attached the paper with further comments. 

Kindest regards

The English language was satisfactory.

Author Response

Reviewer 4

Concern 1: Fix figure 2. The top line has vertical lines through it.

Response: We would like to thank the reviewer for the valuable comments to improve the quality of the manuscript. Figure 2 is corrected and updated in the manuscript according to the reviewer's suggestion.

Concern 2: Reduce the Font size in Table 1.

Response: The table is converted into a list.

Concern 3: Images are blurry

Response: Figures have been improved.

Round 2

Reviewer 1 Report

1.      There is no highlight in the revised manuscript. It is very difficult to find the revised sections.

2.      I don’t think the authors answer the question:

It is not necessary to have two and half pages for literature review. You can have one or two paragraphs for the literature review, but the most important is the limitations of the previous studies, and the novelty of your study.

3.      The authors need to add the answer of question “What is the point of having figure 7? Why the 3-d or 2-d data distribution is important?” to your manuscript.

4.      Only replying to the questions is not enough. The authors need to add the corresponding answers to your revised manuscript, and highlight it. For example, why you mentioned 1024, 512 and 215 for ANN? Please check all the questions.

1.      For the questions: Figure 8 is redundant, because you already have Table3.

                               Figure 10 is redundant, because you already have Table 4.

The results only include several numbers, and they are close to each other. I believe that readers can imagine all the figures.

Author Response

(Reviewer 1)

Concern 1: There is no highlight in the revised manuscript. It is very difficult to find the revised sections.

Response: We would like to thank the reviewer for showing concern which helped us to improve the quality of the manuscript. We highlight changes according to reviewer suggestions in the updated manuscript. We upload the highlighted file as a supplementary file

Concern 2: I don’t think the authors answer the question: It is not necessary to have two and half pages for literature review. You can have one or two paragraphs for the literature review, but the most important is the limitations of the previous studies, and the novelty of your study.

Response: Limitations of studies are discussed in the last paragraph of the literature review. Also, a comparison of the existing study with previous studies is added.

Changes can be found in the updated version of the manuscript and also in the highlighted file in yellow color.

Concern 3:  The authors need to add the answer of question “What is the point of having figure 7? Why the 3-d or 2-d data distribution is important?” to your manuscript.

Response: The purpose of Figure 7 is to provide a comprehensive understanding of the data's arrangement. The visualization facilitates the evaluation of how the data points are dispersed and whether or not they exhibit distinct patterns or separability. By analysing Figure 7 (b), we can see that the data points exhibit a distinct linear separation. The choice between a three-dimensional and a two-dimensional representation enables the development of distinct insights. The three-dimensional perspective provides a holistic view of data distribution, whereas the two-dimensional projection emphasizes the linear relationships within the data. Understanding the dimensionality and separability of the data is essential for selecting suitable classification methods and determining the potential accuracy of linear classifiers in categorizing the data points.

We also answer this concern in the manuscript. Changes can be found in the updated version of the manuscript and also in the highlighted file in yellow color.

Concern 4: Only replying to the questions is not enough. The authors need to add the corresponding answers to your revised manuscript, and highlight it. For example, why you mentioned 1024, 512 and 215 for ANN? Please check all the questions.

Response: The numbers 1024, 512, and 256 correspond to the specific configurations of hidden layers and neuron counts within the Artificial Neural Network (ANN) architecture in our study. These values indicate the number of neurons present in each hidden layer of the ANN. We added this response in the updated manuscript highlighted with yellow color

Concern 5: For the questions: Figure 8 is redundant, because you already have Table3.   Figure 10 is redundant, because you already have Table 4. The results only include several numbers, and they are close to each other. I believe that readers can imagine all the figures.

Response: We appreciate the review suggestion to improve the quality of the manuscript we remove figures which have their corresponding table to reduce the redundancy. Changes can be found in updated manuscripts.

Round 3

Reviewer 1 Report

NA